# MTSegNet:Semi-supervised Abdominal Organ Segmentation in CT

Shiman Li[0000−0002−0488−0573] and Siqi Yin[0000−0003−0012−0187]

Digital Medical Research Center, School of Basic Medical Sciences,
Fudan University,Shanghai 200032, China
{21111010029,21111010030}@m.fudan.edu.cn

**Abstract.** Multi-organ segmentation from CT scan is useful in clinical applications. However, difficulties in data annotation impede its practical usage. In this work, we propose MTSegNet for multi-organ segmentation task in semi-supervised way. Total number of 13 organs in chest and abdomen are included. For network architecture, Attention U-Net serves as basic structure to guarantee segmentation performance and usage of context information. For those unlabeled data, Mean Teacher Model, which is a commonly used semi-supervised structure, is added to the pipeline to facilitate better use of unlabeled data. Besides, class-aware weight and post-process are used as auxiliary methods to further improve performance of model. Experiments on validation set and test set got averaged Dice Similarity Coefficient (DSC) of 0.6743 and 0.7034, respectively.

**Keywords:** Semi-supervised · Multi-organ · Abdominal Segmentation .

## 1   Introduction

Multi-organ segmentation from CT scan has many important practical applications in clinical scene. However, due to manually annotation is time-consuming and labor-intensive, supervised methods are no longer satisfied. In this paper, we focus on semi-supervised learning method for multi-organ segmentation task. There are several challenges. 1) Labels includes 13 organs in chest and abdomen, which has varying size, shape and contour. 2) Besides, according to human body internal structure, organs have complicated context relationship, without regard to its normal and abnormal conditions. 3) Additionally, a large proportion of unlabeled data requires a better way to get fully use of it. 4) The trade-off between model structure and limited GPU memory size.

In this work, we propose a semi-supervised multi-organ segmentation model MTSegNet to effectively and efficiently tackle challenges mentioned above. Attention U-Net serves as basic structure to guarantee segmentation performance and usage of context information. For those unlabeled data, Mean Teacher Model, which is a commonly used semi-supervised structure, is added to the pipeline to facilitate better use of unlabeled data. Besides, class-aware weight and post-process are used as auxiliary methods to further improve performance of model.

The main contributions of this work are summarized as follows:

1. We propose MTSegNet for 13 labels multi-organ segmentation task using CT scans, which based on Attetion U-Net's attention and statistics information to exploit contextual information in a better way.
2. We designs Mean Teacher's consistency structure to make fully use of the given unlabeled data. An auxiliary class-aware weight are setting to abridge the differences of labels
3. Post-process are given to improve the performance of model further, based on statistic prior information.

## 2      Method

For the task of segmenting 13 organs of interest in the abdomen of FLARE2022, we propose MTSegNet, which is based on Mean Teacher's [8] semi-supervised method to segment organs using Attention U-Net on sliding patches. The details of the method will be described as follows.

### 2.1    Preprocessing

The baseline method includes the following pre-processing strategies:

- *Reorienting images.* Reordering original images' direction to left-posterior-inferior view.
- *Resampling images* Two options are given. We use shape resampling in our work.
    - Resampling by space. Resampling image to 1mm spacing in each axis to increase the comparability of images and restore the real physical locations.
    - Resampling by size. Resize images to pre-defined size of inputs (i.e. [160, 160, 160].
- *Intensity normalization* . The image in normalized by specific values of window width and window level, which is 400 and 40, respectively.

### 2.2    Proposed Method

Our proposed MTSegNet contains three effective methods: Mean Teacher's consistency regularization, Attetion U-Net's attention mechanism, and class-aware weight setting.

The main framework of our approach is shown in Fig. 1 We input labelled data into the student network and unlabeled data into the student and teacher networks at the same time. We averages the weights of the student model to the teacher model, which ensuring the stability of the model.

Among them, we use Dice loss and cross entropy loss for the labeled images, because compound loss functions have been proved to be robust in various medical image segmentation [5]. In order to supervise the unlabeled data, we calculate the consistency loss using the prediction results of the teacher and student model for regularization as a way to improve the model generalization.

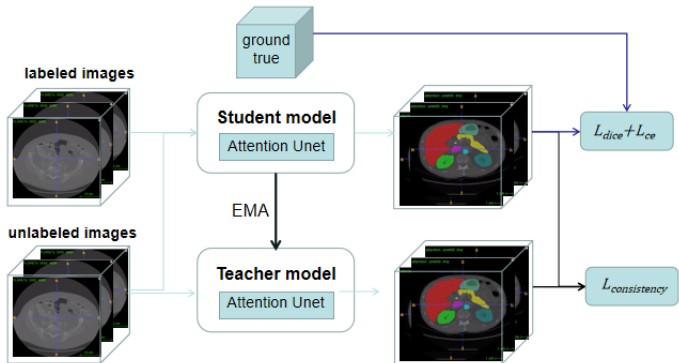

**Fig. 1.** Network architecture

For the segmentation of multi-organs, there are often many irrelevant regions in the image for the segmentation of one type of organs, and the resulting redundant information will affect the segmentation results. Therefore, we use the Attention U-Net as the basic backbone network, and its Attention Gate attention mechanism can help the network learn the spatial location and other features of different organs .

In addition, the segmentation difficulty of different organs varies, and in order to better improve the overall segmentation results, we set class-aware weights and set higher weights for the more difficult segmented organs in calculating the loss to help the network learn the difficult organs for segmentation. The class-aware weights is set as a super-parameter, which is manually adjusted before training. The specific implementation is in the form of multiplication of loss and class weights. We set the weight as 1.0, 1.0, 1.0, 1.0, 1.5, 1.0, 1.5, 2.0, 2.0, 1.5, 2.0, 2.0, 2.0, 1.0 for background and classes 1 to 13. The Dice loss is formulated as follow:

$$Loss_{Dice} = 1 - \sum_{i=0}^{c=13} \frac{2\,|P_i \cap Y_i|}{|P_i| + |Y_i|} \times W_i \tag{1}$$

where, $P_i$ and $Y_i$ indicate the prediction and ground truth of class $i$, respectively. And $W_i$ indicates the weight of the class. Based on the solution of nnUnet [4], we recommend using Float16 as the tensor type to improve inference speed and reduce resource consumption.

### 2.3 Post-processing

To refine the segmentation output, several post-processing methods are considered.

**Statistic information** Since basic structure of human body is similar between individuals, we stat the volume, location information of each label in the given labeled cases (after pro-processing) to make fully use of available data.

Figure 2 shows labels location information after uniformly resized to same size. For each label, box plot includes six boxes corresponding to minimal and maximal coordinates along z/y/x axis, which reflects a general locations and deviation of labels in labeled cases. Histogram in figure 3 shows each labels volume statistics. All the information collected from labeled data may utilize as a reference for incoming unlabeled data in post-processing process.

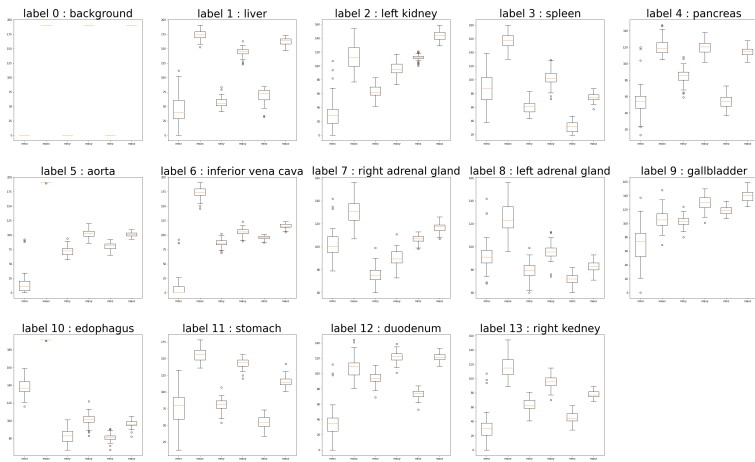

**Fig. 2.** Labels location statistics

Based on statistics information, post-processing includes two steps: First, Connected Component Analysis-Labeling is applied on the model output. Additionally, some small isolated prediction regions are removed based on the statistic information on each labels.

## 3   Experiments

### 3.1   Dataset and evaluation measures

Description of dataset : The FLARE2022 dataset is curated from more than 20 medical groups under the license permission, including MSD [7], KiTS [2,3], AbdomenCT-1K [6], and TCIA [1].

Description of data : The training set includes 50 labelled CT scans with pancreas disease and 2000 unlabelled CT scans with liver, kidney, spleen, or pancreas diseases. The validation set includes 50 CT scans with liver, kidney, spleen, or pancreas diseases. The testing set includes 200 CT scans where 100 cases has liver, kidney, spleen, or pancreas diseases and the other 100 cases has uterine corpus endometrial, urothelial bladder, stomach, sarcomas, or ovarian diseases. All the CT scans only have image information and the center information is not available.

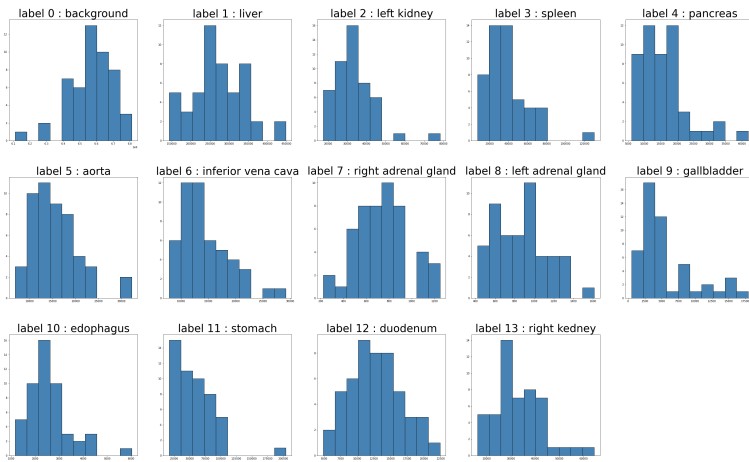

**Fig. 3.** Labels volume statistics

The total number of cases is 50 labeled data and 2000 unlabeled data. K-fold (k=5) training strategy are used for labeled data for train/validate/test splitting. Unlabeled data are chosen for training only.

The evaluation measures consist of two accuracy measures: Dice Similarity Coefficient (DSC) and Normalized Surface Dice (NSD), and three running efficiency measures: running time, area under GPU memory-time curve, and area under CPU utilization-time curve. All measures will be used to compute the ranking. Moreover, the GPU memory consumption has a 2 GB tolerance.

### 3.2 Implementation details

**Environment settings** The environments and requirements are presented in Table 1.

**Table 1.** Environments and requirements.

| Windows/Ubuntu version | Ubuntu 20.04 |
|---|---|
| CPU | Intel(R) Core(TM) i5-9600K CPU @ 3.70GHz |
| GPU (number and type) | Nvidia GeForce 2080Ti (×2) |
| CUDA version | 11.4 |
| Programming language | Python 3.6 |
| Deep learning framework | Pytorch (Torch 1.10.2, torchvision 0.4.0) |

**Training protocols** As for the training protocols, we introduce some details by the form of table as follow:

**Table 2.** Training protocols.

| | |
|---|---|
| Network initialization | "he" normal initialization |
| Batch size | 4 |
| labelled number in Batch size | 2 |
| Patch size | 96×96×96 |
| stride x/y/z | 48×48×48 |
| Total epochs | 1000 |
| Optimizer | SGD with nesterov momentum ($\mu = 0.9$) |
| Initial learning rate (lr) | 0.01 |
| Lr decay schedule | linear decay |
| Training time | 14-16 hours |
| Number of model parameters | 6.17M[1] |
| Number of flops | 59.02G[2] |

To be more specific, the patch sampling strategy in the training phase is the random crop, while in the validating phase we slide the patch by the stride in the table.

## 4    Results and discussion

Unlabeled data are used in Mean Teacher Model and consistency loss calculation. 2000 cases of unlabeled data ease the problem of annotation lacking and could help model to be more robust.

To further improve model performance, class-aware weight are added, which balance the internal differences between labels to a great extent. Meanwhile, Attention U-Net has the ability to focus on context information and make fully use of anatomy structural locations.

However, dice score and some measurements are not satisfied now, possible reasons are: 1) unsuitable input patch size, which may mislead model with inadequate input information. 2) abnormal cases may confuse model. How to identify special cases and let the model rely more on high confidence cases are remain unsolved.

### 4.1    Quantitative results on validation set

We randomly divide the 50 labeled data into training and validation sets in a 4:1 ratio and use the k-fold approach for validation. Result are shown in table 3.

The use of unsupervised data improves the performance of the model. However, with the modification of the model, we lost the original Ablation Experiment on the effect of unlabeled data, and no new ablation experiment has been carried out.

**Table 3.** Performance on validation set.

| organ | Liver | RK | Spleen | Pancreas | Aorta | IVC | RAG |
|-------|-------|------|--------|----------|-------|------|------|
| DSC | 0.9202 | 0.8149 | 0.8298 | 0.611 | 0.8673 | 0.7783 | 0.4905 |
| organ | LAG | Gallbladder | Esophagus | Stomach | Duodenum | LK | Mean |
| DSC | 0.4033 | 0.4162 | 0.6432 | 0.7462 | 0.5046 | 0.7406 | 0.6743 |

Firstly, table 4 show the effectiveness of preprocessing on the training dataset, which lead to better evaluation result in most labels, especially for pancreas, right adrenal gland and left adrenal gland.

Besides, we explore the effectiveness of class-aware weight on training data in table 5. Similar to the effect of preprocessing, most labels' measurement has increased.

**Table 4.** Ablation study of preprocess.

| | label 1 | | label 2 | | label 3 | | label 4 | | label 5 | | label 6 | | label 7 | |
|---|---|---|---|---|---|---|---|---|---|---|---|---|---|---|
| | dice_score | hd95 | dice_score | hd95 | dice_score | hd95 | dice_score | hd95 | dice_score | hd95 | dice_score | hd95 | dice_score | hd95 |
| w/o preprocess | 0.9716 | 1.5890 | 0.8684 | 10.0501 | 0.9505 | 2.8849 | 0.7255 | 5.4089 | 0.9381 | 3.9550 | 0.8387 | 3.6709 | 0.7350 | 2.0953 |
| preprocess | 0.9690 | 3.6099 | 0.8714 | 10.9476 | 0.9271 | 3.8832 | 0.7749 | 4.5959 | 0.9321 | 1.2472 | 0.8618 | 4.0251 | 0.8168 | 1.4988 |
| | label 8 | | label 9 | | label 10 | | label 11 | | label 12 | | label 13 | | Average | |
| | dice_score | hd95 | dice_score | hd95 | dice_score | hd95 | dice_score | hd95 | dice_score | hd95 | dice_score | hd95 | dice_score | hd95 |
| w/o preprocess | 0.6757 | 5.8350 | 0.8731 | 7.5368 | 0.7629 | 2.6278 | 0.8562 | 6.2145 | 0.7221 | 6.0233 | 0.9276 | 6.5980 | 0.8343 | 4.9607 |
| preprocess | 0.7332 | 3.2363 | 0.8937 | 4.3394 | 0.7243 | 4.4726 | 0.8825 | 8.1856 | 0.7217 | 5.7623 | 0.9461 | 10.1753 | 0.8503 | 5.0753 |

**Table 5.** Ablation study of class-aware weight.

| | label 1 | | label 2 | | label 3 | | label 4 | | label 5 | | label 6 | | label 7 | |
|---|---|---|---|---|---|---|---|---|---|---|---|---|---|---|
| | dice_score | hd95 | dice_score | hd95 | dice_score | hd95 | dice_score | hd95 | dice_score | hd95 | dice_score | hd95 | dice_score | hd95 |
| w/o class-aware weight | 0.9624 | 5.0721 | 0.8659 | 9.8311 | 0.9330 | 3.1310 | 0.7246 | 4.7754 | 0.9232 | 4.6414 | 0.8204 | 3.8115 | 0.7173 | 4.6115 |
| class-aware weight | 0.9716 | 1.5890 | 0.8684 | 10.0501 | 0.9505 | 2.8849 | 0.7255 | 5.4089 | 0.9381 | 3.9550 | 0.8387 | 3.6709 | 0.7350 | 2.0953 |
| | label 8 | | label 9 | | label 10 | | label 11 | | label 12 | | label 13 | | Average | |
| | dice_score | hd95 | dice_score | hd95 | dice_score | hd95 | dice_score | hd95 | dice_score | hd95 | dice_score | hd95 | dice_score | hd95 |
| w/o class-aware weight | 0.7220 | 4.5190 | 0.8764 | 2.4145 | 0.7748 | 2.3610 | 0.8300 | 8.1811 | 0.6284 | 6.3779 | 0.9206 | 8.8114 | 0.8230 | 5.2722 |
| class-aware weight | 0.6757 | 5.8350 | 0.8731 | 7.5368 | 0.7629 | 2.6278 | 0.8562 | 6.2145 | 0.7221 | 6.0233 | 0.9276 | 6.5980 | 0.8343 | 4.9607 |

### 4.2 Visualization

To give an further analyze to the result in an more intuitive way, we visualize several cases, including its prediction mask and ground truth label.

Fig. 4 shows some examples with satisfactory result. Due to the quantity of labels, We visualize each case from multi-view as show in the first three columns. Tubular lumen-like organ (i.e. Aorta and IVC) has considerably good result, which shows the effectiveness of attention module. When observing the details

of predictions, the edge and surface is not as smooth as the ground truth, indicating that a powerful post-processing method is needed to further improve the performance.

Fig. 5 shows several cases with unsatisfactory result. The possible reason of the performance degradation is that the original input has large range in z-axis. In another words, ground truth labels are located in just few slices, which may introduce nuisance information and increase the difficulties for trained network to predict the coarse location of organs and its final segmentation mask. Thus, although the main slices have ground truth-like prediction, those extra area shows in the lower part (i.e. hip) are definitely wrong. To reduce the impact of input size, a location network or post-process method may be helpful to give a coarse suggestions on alternative slices, or re-fine the prediction result through prior knowledge of anatomy.

Unknown disease is also challenging, as it increase the variation of organs between cases. Fig. 6 shows several failed prediction, which possibly caused by abnormal organs, such as liver in validate case 0044 and left kidney in validate case 0023. Moreover, organ like stomach has a disappointed result as the anatomical variability are comparably large between cases, such as validate case 0003 and validate 0012 shown in fig. 4. Similarly, pancreas is easy to be misclassified, due to its contrast and variable shape. Since 14 organs has mutual constraints, mining relative position between organs and anatomical structure should be considered in the future.

### 4.3   Quantitative results on test set

Results on test set are shown in table 6. Including average score (AVG) and standard deviation (STD) of DSC and NSD of each organ and final mean result of all

**Table 6.** Performance on test set.

|     | Name | Liver | RK | Spleen | Pancreas | Aorta | IVC | RAG |
|-----|------|-------|-----|--------|----------|-------|-----|-----|
| DSC | AVG | 0.9483505 | 0.8618665 | 0.875381 | 0.5917635 | 0.9080215 | 0.825371 | 0.524265 |
|     | STD | 0.037949668 | 0.207216528 | 0.206019203 | 0.241541075 | 0.086333642 | 0.133338821 | 0.254396351 |
|     | | LAG | Gallbladder | Esophagus | Stomach | Duodenum | LK | Mean |
| DSC | AVG | 0.4540385 | 0.43166 | 0.578257 | 0.770988 | 0.527637 | 0.8462805 | 0.703375385 |
|     | STD | 0.288194299 | 0.3657501 | 0.24846958 | 0.170715222 | 0.214083741 | 0.198379682 | 0.204029839 |
|     | | Liver | RK | Spleen | Pancreas | Aorta | IVC | RAG |
| NSD | AVG | 0.904239 | 0.8239195 | 0.8553155 | 0.71848 | 0.9305165 | 0.818754 | 0.6904825 |
|     | STD | 0.100097309 | 0.224250366 | 0.217654229 | 0.242036863 | 0.109057073 | 0.148377794 | 0.282271853 |
|     | | LAG | Gallbladder | Esophagus | Stomach | Duodenum | LK | Mean |
| NSD | AVG | 0.592652 | 0.3834735 | 0.712121 | 0.7552445 | 0.755684 | 0.7963375 | 0.749016885 |
|     | STD | 0.328952015 | 0.330597739 | 0.264299128 | 0.185882842 | 0.199104964 | 0.211049075 | 0.218740865 |

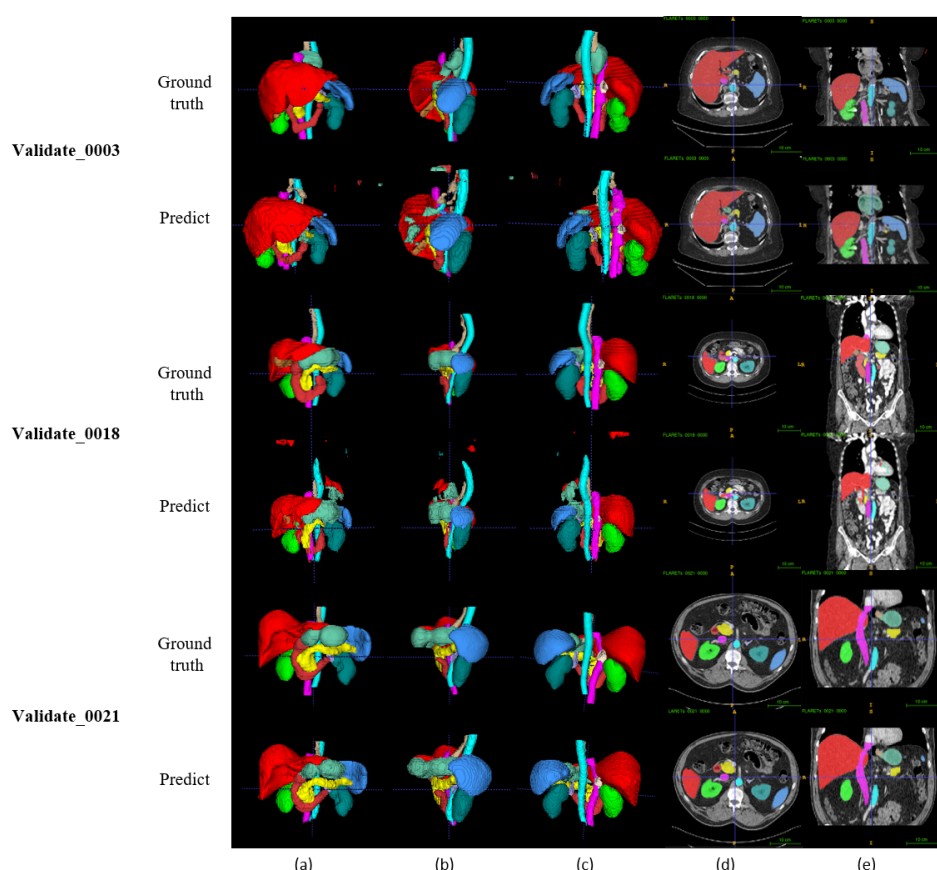

**Fig. 4.** Visualization of cases with satisfying result. (a)Front view. (b)Side view. (c)Back view. (d)Axial view. (E)Coronal view.

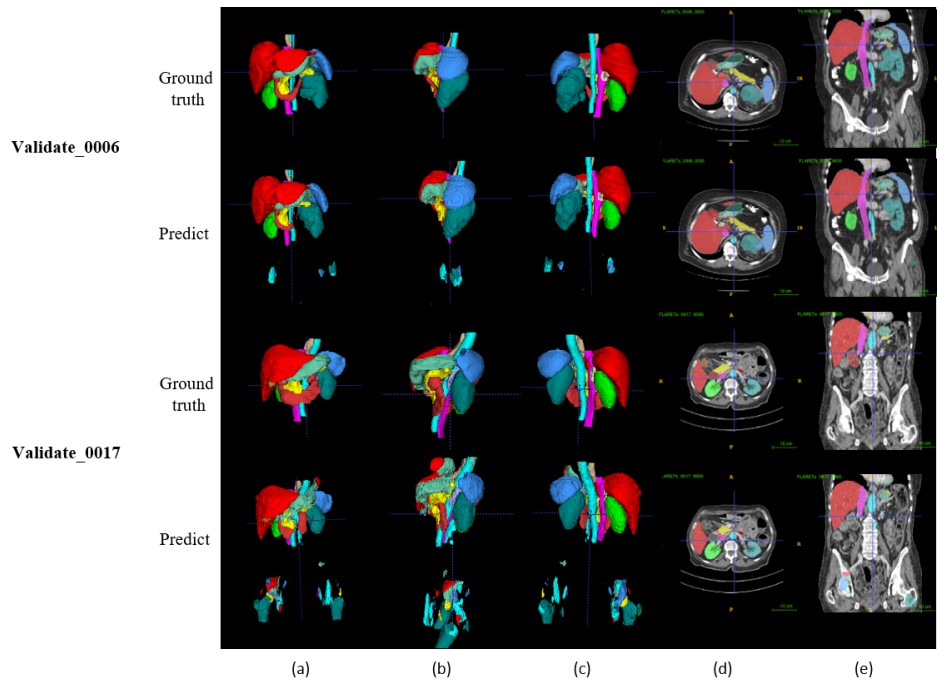

**Fig. 5.** Visualization of Unsatisfactory results. (a)Front view. (b)Side view. (c)Back view. (d)Axial view. (E)Coronal view

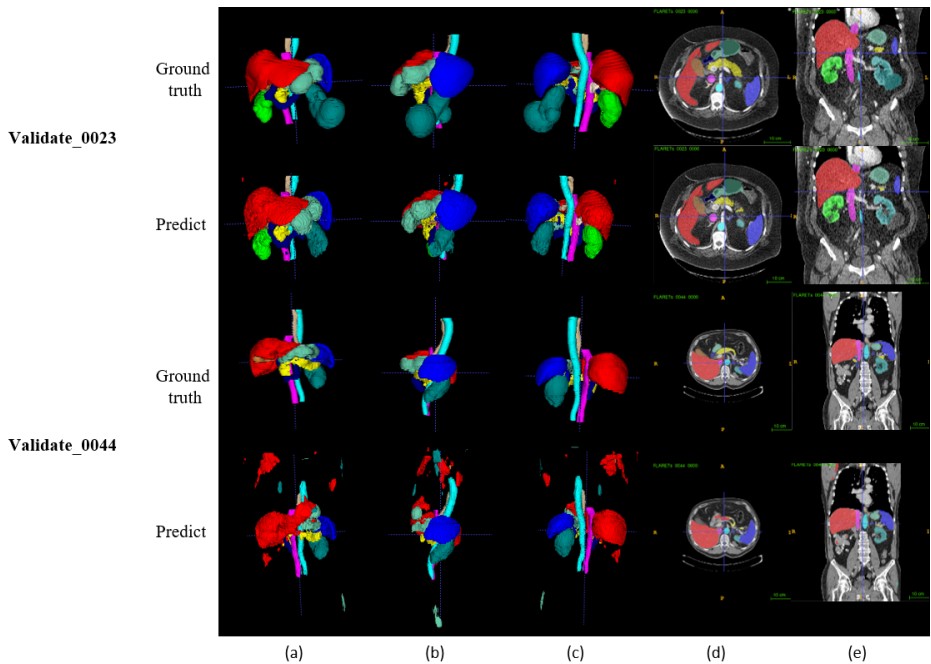

**Fig. 6.** Visualization of difficult cases. (a)Front view. (b)Side view. (c)Back view. (d)Axial view. (E)Coronal view

## 5   Conclusion

The main finding and results show that, MTSegNet shows its ability in multi-organ segmentation tasks. Both labeled and unlabeled data contributes to model training by using Mean Teacher model and Attention U-Net Model. Besides, preprocessing and class-aware weight helps further improvement in model performance. However, there are still many drawbacks need to be completed in the future.

**Acknowledgements** The authors of this paper declare that the segmentation method they implemented for participation in the FLARE 2022 challenge has not used any pre-trained models nor additional datasets other than those provided by the organizers.

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
