# OpenReview forum: "MTSegNetSemi-supervised Abdominal Organ Segmentation in CT"
_MICCAI.org/2022/Challenge/FLARE_

### Official Review · Reviewer_dQUK · 2022-09-16
**Good work but talk about the network structure little.**

**Rating:** 7
**Confidence:** 3

**Review:**

Pros:
1)In this work, they take the mean teacher method to use the unlabeled cases.
2)They study the class-aware weight, preprocessing  and postprocessing influence
3) The final mean DSC is 0.8503.
Cons:
1) talk little about the details of the network.
2) missing the experiment of unlabled cases.
3) It will be better to give the formula of the class-aware weight.

---

### Official Review · Reviewer_KxLe · 2022-09-16
**No statistics for the inference efficiency and resource utilization such as GPU and CPU**

**Rating:** 8
**Confidence:** 3

**Review:**

Pros: Unlabeled data is well utilized by using the Mean Teacher Model.

Cons: No statistics for the inference efficiency and resource utilization such as GPU and CPU.

---

### Official Review · Reviewer_yr3z · 2022-09-19
**Review of MTSegNetSemi-supervised Abdominal Organ Segmentation in CT**

**Rating:** 8
**Confidence:** 4

**Review:**

The authors give clear and detailed description of their method and experiments.

---

### Meta-Review · Program_Chairs · 2022-09-28

**Recommendation:** Major Revision
**Confidence:** 5

**Metareview:**

Reviewers raise many concerns and suggestions. Please address all comments in the revised manuscript.